# Importance of Host Feeding in the Biological Control of Insect Pests: Case Study of Egg Parasitoid Species (Hymenoptera: Chalcidoidea: Trichogrammatidae)

**DOI:** 10.3390/insects15070496

**Published:** 2024-07-03

**Authors:** Tomas Cabello, Juan Ramón Gallego, Inmaculada Lopez, Manuel Gamez, Jozsef Garay

**Affiliations:** 1Centre for Agribusiness Biotechnology Research, University of Almería, La Cañada de S. Urbano, s/n, ES-04120 Almería, Spain; jgg436@ual.es; 2Department of Mathematics, University of Almería, La Cañada de S. Urbano, s/n, ES-04120 Almería, Spain; milopez@ual.es (I.L.); mgamez@ual.es (M.G.); 3HUN-REN Centre for Ecological Research, Institute of Evolution, Konkoly-Thege Miklosut 29–33, H-1121 Budapest, Hungary; garay.jozsef@ecolres.hu

**Keywords:** intraguild predation, facultative hyperparasitoid, functional response, synovigeny, *Trichogramma achaeae*, *Trichogramma brassicae*

## Abstract

**Simple Summary:**

The use of oophagous species from the genus *Trichogramma*—small wasp—in the control of pest species, both in agriculture and forestry, is the most important example of this type of control because of the area in which it is implemented. They are called parasitoids because the female seeks out the host—the pest species—from which its offspring feed and develop. Until now, this mechanism of action was thought to be due to the parasitism relationships established between the immature stages of the parasitoid and its host; however, it has been shown that the directed ‘host-feeding’ mechanism of the adult mothers, acting as predators, plays a very important ecological role. This proves that there are much more complex parasitoid–parasitoid relationships than predator–predator relationships. This has implications for the current state of knowledge and can be applied in biological control involving parasitoid species.

**Abstract:**

Over recent decades, intraguild predation (IGP) has attracted special attention, both from the theoretical and practical standpoints. The present paper addresses the interference competition between two *Trichogramma* species (egg parasitoids)—on the one hand, the extrinsic interactions (i.e., the indirect competition between female *T. achaeae* and *T. brassicae*), and on the other, the intrinsic interactions between the larvae of both species. Furthermore, *T. achaeae* is a better competitor than *T. brassicae* due to a dual mechanism—the former acts as a facultative hyperparasitoid of the latter, exclusively considering parasitism relationships as well as presenting predation activity by host feeding, which gives preference to eggs previously parasitized by *T. brassicae* over non-parasitized eggs. Both mechanisms are dependent on the prey density, which is demonstrated by a change in the functional response (i.e., the relationship between the numbers of prey attacked at different prey densities) of *T. achaeae* adult female—it changes from type II (i.e., initial phase in which the number of attacked targets increases hyperbolically and then reaches an asymptote, reflecting the handling capacity of the predator), in the absence of competition (an instantaneous search rate of *a*′ = 9.996 ± 4.973 days^−1^ and a handling time of *T_h_* = 0.018 ± 0.001 days), to type I (i.e., linear increase in parasitism rate as host densities rise, until reaching a maximum parasitism rate, and an instantaneous search rate of *a*′ = 0.879 ± 0.072 days^−1^ and a handling time of *T_h_* ≈ 0) when interference competition is present. These results show that there is a greater mortality potential of this species, *T. achaeae*, in conditions of competition with other species, *T. brassicae* in this case. Based on this, their implications in relation to the biological control of pests by parasitoid species are discussed.

## 1. Introduction

Parasitoids are parasitic insects whose larvae develop by feeding on or within arthropod hosts [1]. A single host individual sustains the development of one or more parasitoids and the host is almost invariably killed by the interaction [2]. Most species (e.g., [3]). The average number of parasitoid species per host insect species is five, but the total range observed is from zero to 50 or more [4]. In addition to the host range width, parasitoid guilds can be defined by the developmental stage at which the host is attacked by the parasitoid and the mode of parasitism (i.e., ectoparasitic or endoparasitic) [2,4]. 

At the guild level, and under certain conditions, multiple species of parasitoids may compete for host resources [5,6]. Parasitized hosts represent a fixed food supply, and, with rare exceptions, they cannot move from one host to another [7].

Competition in parasitoids can be divided into intrinsic or extrinsic competition [8,9]. Extrinsic competition among free-living parasitoid adults searching for host resources occurs when an adult female reaches a host patch that has been previously parasitized and reacts to it, usually by changing her progeny, sex allocation, and patch residence time [5,10]. Intrinsic competition occurs among immature parasitoids developing on or inside the host when two or more adult females of the same or distinct parasitoid species simultaneously exploit a host patch and then their larvae compete for the same host [11,12]. This can lead to cases of superparasitism/multiparasitism (when more parasitoids of the same/different species are developing on or within a host insect than can survive to maturity) or facultative hyperparasitism (when the parasitoid can complete feeding and development as a primary parasite or use a primary parasitoid as a host) [5,10].

With regard to intraguild predation (IGP) in parasitoid species, competition and parasitism have received preferential treatment, whereas the predatory behavior of host feeding by adult females has been overlooked. The process involves parasitoid females obtaining nutrients from the host by feeding on the body fluids that exude from wounds inflicted by the ovipositor [13,14,15]. Host feeding is important for parasitoid fecundity [16,17]. This may be especially important in parasitoid species that are synovigenic (they continue to mature eggs during their adult phase) versus species that are proovigenic (they have all their eggs ready for oviposition at emergence and do not produce more during their lifetime). Egg parasitoid species are currently considered to be moderately synovigenic [18]. Host-feeding behavior has been recorded in females from 17 different families of Hymenoptera [19] and 2 of Diptera [20]; moreover, parasitoid species kill significant numbers of hosts by host feeding as well as by parasitism, which is important for reducing the population size of insect pests [21,22,23]. Nevertheless, no experimental work has been conducted that assesses the importance of IGP in terms of host feeding [24].

Following the publication of a general IGP theory by Holt and Polis [25], many theoretical and empirical studies have highlighted the importance of functional responses in these types of interactions (e.g., ([26,27,28,29]). The term ‘functional response’ was originally coined by Solomon [30] to describe the way that prey density affects the number of preys attacked per predator per unit time [31]. Furthermore, the functional response has been used extensively in population ecology and foraging theory to study the potential of natural enemies to regulate host populations [32,33]. The functional response type (i.e., type I, type II, or type III) is characterized by the shape of the curve that describes the attack rates, which are influenced by handling times (*T_h_*) and the search efficiencies (*a*′) of natural enemies over increasing host densities within a fixed exposure time [34,35,36]. However, no empirical study has simultaneously examined the form of the functional response or the effect of prey density on IGP, with the exceptions of the works by Kestrup et al. [37] and Sentis et al. [38], which were only on predatory species. 

To clarify and delve more deeply into the above-mentioned aspects, this work has been carried out on two species. The Trichogrammatidae family includes 63 genera within the Chalcidoidea superfamily. *Trichogramma* species, all of which are exceedingly small insects (0.2–1.5 mm), are either solitary or gregarious endoparasitoids of insect eggs, mainly lepidopteran but also dipteran, coleopteran, neuropteran, and hymenopteran. They develop inside the host egg as idiobiont parasitoids (i.e., the parasitoid females kill at once, or shortly after initial parasitization, permanently impairing or preventing further development of the hosts) [4]. *Trichogramma* species are applied as biological control agents, primarily by inundative release, in more than 50 countries and on more than 32 million hectares of both agricultural and forest land. These parasitoid eggs mainly control lepidopteran pest species in crops such as maize, cotton, sorghum, soybean, sugarcane, tomato, and grapevine [39,40].

In this work, we focus on providing more information on the importance of host feeding, specifically as follows:(1)To evaluate the importance of host feeding in the parasitoid–parasitoid IGP;(2)To check if there are different host-feeding behaviors exhibited by adult females between taxonomically close species and to assess their impact on intraspecific competition; and(3)To study whether host-feeding mortality depends on pest density when there is parasitoid–parasitoid IGP.

## 2. Materials and Methods

Three different sets of trials were performed. In the first set, comprising only one trial, indirect (extrinsic) competition and direct (intrinsic) competition were evaluated in the two *Trichogramma* species at a low host density. In the second set, we evaluated the functional responses of adult females of both species (two trials). Finally, in the last set (two trials), we also evaluated the functional responses but in terms of interference competition.

All the trials were carried out at 25 ± 1 °C and 60–80% R.H., and 16:8 h of light: darkness under laboratory conditions.

### 2.1. Biological Material

The insects used in the different trials were obtained from colonies bred in a laboratory (Agricultural Entomology, University of Almeria, Almeria, Spain). Both species of *Trichogramma* were reared on *Ephestia kuehniella* eggs. The laboratory rearing of the three species was carried out according to the method described by Parra et al. [41].

### 2.2. Experimental Design and Procedures

#### 2.2.1. Experiment 1: *T. achaeae*–*T. brassicae* Interaction

The trial had a single treatment at five levels, including the control. Ten replications were performed for each treatment, each of them consisting of a batch of 5 host eggs.

The method of Cabello et al. [42] was followed, in which 24- to 48-h-old adult females of each parasitoid species were used for testing. The *Trichogramma* female adults were obtained by isolating parasitized *E. kuehniella* eggs. This was performed by removing them with a fine wet brush (no. 0). The isolated eggs were individually placed in glass vials (0.9 cm wide and 5.0 cm long) covered with cotton, and subsequently allowed to develop until the adults emerged. Upon emergence, their sex was determined.

Only adult virgin females were used for this trial to ensure only male offspring emerged. The reproduction system of this species is arrhenotokous parthenogenesis (males are haploid while females are diploid). This allowed us to identify the emerging *Trichogramma* species by morphological features, which can only be performed with adult males. Additionally, fecundity in mated females is similar to that of virgin females.

The parasitization containers used were glass vials (1.0 cm wide and 1.5 cm long) in which a white cardboard strip (0.9 cm wide and 5.0 cm long) was placed. On the cardboard, 5 *E. kuehniella* eggs (less than 24 h old) were glued with distilled water using a fine brush. These eggs were previously sterilized by UV irradiation to prevent the emergence of phytophagous larvae. The females were not given any food. 

In each vial, one female from each of the *Trichogramma* species was placed alongside the host eggs, except in the control group. After 24 h, all the females were removed. Subsequently, in the sequential treatments (parasitism by one species and then, by the other), an adult female of the other species was introduced. This female was also left to parasitize the host eggs for 24 h, after which they were also removed (Figure 1). 

During these periods of parasitoid exposure to adult females, each container was observed (in all the treatments and replications) using a stereoscopic microscope for 1 min every 40 min to assure that the eggs were accepted by the females for parasitization and/or host feeding.

After parasitization, all the cardboard strips containing the host eggs were allowed to evolve under the aforementioned conditions until the parasitoid offspring emerged (at about 15 days). The controls were also subjected to the above-mentioned procedure but not exposed to *Trichogramma* female parasitization. All the treatments (10 replicates per treatment) and control (10 replicates) were carried out simultaneously over the test period.

In this trial, the data collected were the number of host eggs killed by parasitism or host feeding as well as the species of the male adults that emerged from the parasitized eggs. In accordance with the method described by Hansen and Jensen [43], the number of dead parasitized and non-parasitized eggs was recorded. Dead parasitized eggs could be discerned from the dead non-parasitized ones by the black color of the former as opposed to the lighter brown of the latter. Furthermore, the former often kept their round shape while dead non-parasitized eggs collapsed after a short time; dead non-parasitized host eggs, adjusted for control mortality, were considered to have been killed by host feeding. 

All the male adults that emerged were fixed in alcohol–glycerol (10%), clarified in lactophenol solution, and slide-mounted in Hoyer’s fluid. The specimens were identified by their morphological characteristics under a light microscope.

The number of parasitized or dead eggs from host feeding was analyzed assuming a binomial distribution for the random component of a generalized linear model (GZLM) [44]. The data were subjected to the GenLin procedure with this binomial distribution and logit link function using IBM SPSS version 22.0 statistical software.

#### 2.2.2. Experiment 2: Functional Response of *T. achaeae* and *T. brassicae*

Two trials were carried out. The first determined the functional response of *T. achaeae* and the second that of *T. brassicae*. The trials followed a completely randomized design, with only one factor: prey density at five treatment levels (10, 30, 50, 70, and 90 host eggs).

Egg hosts were used in both testing groups: *E. kuehniella*, less than 24 h old (UV irradiated), and virgin female adults of *T. brassicae* o *T. achaeae*, less than 24 to 48 h old since the emergence of the egg host. The reported method was followed in all cases. 

At each host density, the eggs were attached to the 3 × 1 cm cardboard strips by using a wet brush, isolated in rows (1 mm of separation between the rows and 1 mm between eggs). There were 10 repetitions for each treatment level. Each of the repetitions consisted of a virgin *Trichogramma* female. The exposure time of the hosts to parasitoids was 1 day. The females were not given any food. The adults were removed afterwards, and the cardboard strips were left to evolve until offspring emergence (15 days). The controls (at each host density; 10 replicates) were subjected to the same procedure but were not exposed to parasitization. All the treatments and the control were carried out simultaneously.

Two types of statistical analyses were applied. Firstly, a logistic regression was determined between the ratio of dead host eggs (parasitism + host feeding) and the available host density. This was carried out according to the polynomial function used by Juliano [36]:(1)$$\frac{N_{a}}{N_{0}}=\frac{exp(P_{0}+P_{1}N_{0}+P_{2}N_{0}^{2}+P_{3}N_{0}^{3})}{1+exp(P_{0}+P_{1}N_{0}+P_{2}N_{0}^{2}+P_{3}N_{0}^{3})}$$
where *N_a_* is the number of dead prey, *N*_0_ is the initial value of available prey, and *P*_0_, *P*_1_, *P*_2,_ and P_3_ stand for the cut-off, linear, square, and cubic coefficients, respectively, estimated using the maximum likelihood method. The *P*_0_–*P*_3_ parameters were obtained by logistic regression. The logistic regression procedure and the maximum likelihood method estimation were carried out using the Statgraphic Centurion XVI version 16.1.18 statistical software package. Regarding the results, if the *P*_1_ coefficient was significantly non-different from zero, it represented a type I functional response (it was considered different from zero when the latter was not included in its confidence interval); a significantly negative value of *P*_1_ indicated type II; and a significantly positive value of *P*_1_ indicated type III.

Secondly, the data were adjusted to the three functional response types according to the equations for parasitoid species by Hassell [45] and Cabello et al. [46], as follows:(2)$$\mathrm{Type}\;I\;\;N_{a}=N_{t}\left[1-exp\left(-a^{\prime }\cdot T\cdot P_{t}\right)\right]$$
(3)$$\mathrm{Type}\;\mathrm{II}\;\;N_{a}=N_{t}\left[1-exp\left(-\frac{a^{\prime }\cdot T\cdot P_{t}}{1+a^{\prime }\cdot T_{h}\cdot P_{t}}\right)\right]$$
(4)$$\mathrm{Type}\;\mathrm{III}\;\;N_{a}=N_{t}\left[1-exp\left(-\frac{\alpha \cdot T\cdot {N_{t}\cdot P}_{t}}{1+\alpha \cdot T_{h}\cdot N_{t}+\alpha \cdot T_{h}\cdot N_{t}^{2}}\right)\right]$$
where *N_a_* is the number of dead hosts; *N_t_* is the initial number of available hosts; *a*′ is the instantaneous search rate (equivalent to Nicholson–Bailey’s “area of discovery”: *a* = *a*′·*T*, days^−^^1^); *T* is the total available search time (days), in this case = 1 day; *P_t_* is the number of parasitoids, *P_t_* = 1 in this case; *T_h_* is the handling time (days); and *α* is the potential for host mortality by the parasitoid species (per unit). The latter is necessary because the *a*′ value is not constant for the type III functional response. Please refer to Equation (5) for the equivalence between *a*′ and *α* according to Cabello et al. [46]:(5)$$a^{\prime }=\frac{\alpha \cdot N_{t}}{1+\alpha \cdot T_{h}\cdot N_{t}}$$
where *a*′, *N_t_*, *α*, and *T_h_* are as before.

Subsequently, the data were analyzed, with Equations (2)–(4), employing Marquardt’s algorithm [47] to fit the non-linear regressions with the Tablecurve 2D statistical software package (version 5.0). To choose the most significant equations, we used the corrected Akaike criterion (AICc) [47,48].

#### 2.2.3. Experiment 3: Effect of *T. achaeae*–*T. brassicae* Interaction on the Functional Response

A split-plot design was used in each trial with a principal factor: density of hosts, at five levels: 10, 30, 50, 70, and 90 host eggs and the secondary factor was the interaction, at two levels: prior parasitization (or not) by the other parasitoid species (half and half host eggs). There were 10 replicates per treatment at each trial.

For previous parasitization, half of the eggs at each density (5, 15, 25, 35, and 45 eggs) were attached to cardboard strips using a wet brush and offered (for 24 h) to virgin females of a species different to the one that was the subject of the trial. The parasitism rate used was the same as in Test 1 (1 adult female/5 host eggs). After this period, the females were removed, and the corresponding densities were completed by attaching new non-parasitized host eggs by a wet brush. These eggs were placed in paired columns alongside the earlier ones. Controls for each treatment and trial were subjected to the above-mentioned procedure; these were not exposed to parasitization by the second *Trichogramma* species.

The data for the dead host eggs (parasitism + host feeding) were fitted to the three functional response types according to the previously described method. Furthermore, the host-feeding mortality data were analyzed by GZLM using the procedure and statistical software mentioned above.

## 3. Results

### 3.1. T. achaeae–T. brassicae Interaction

Nearly all host eggs were killed, either by parasitism or host feeding. There was no difference between eggs parasitized by each *Trichogramma* species alone or sequentially, except for the control group (Figure 2) (Omnibus test, likelihood ratio χ^2^ = 2.704, d.f. = 3; *P* = 0.44). Adult females of both species accepted and laid eggs on non-parasitized host eggs (90.0 ± 4.2 and 90 ± 4.2% of parasitized host by *T. achaeae* and *T. brassicae*, respectively) as well as eggs that had previously been parasitized by the other species (88.0 ± 4.6 or 36.0 ± 7.1, and 49.0 ± 7.5 or 4.0 ± 2.8% of parasitized host by *T. achaeae* and *T. brassicae*, respectively) (Figure 2).

However, (Figure 3) when *T. achaeae* females parasitized first, the emerged adults consisted of 96.3 ± 3.9% of this species, significantly higher than the 3.67 ± 2.46 % of the other one. When the *T. brassicae* parasitized first, the values were closer to each other, but with a significantly higher value for species (55.33 ± 6.00 %) than for the second (44.67 ± 6.01 %).

In contrast, the percentage of eggs killed by host feeding showed significant effects. In the GZLM analysis, the fitted model of dead eggs by host feeding was statistically significant (Omnibus test, likelihood ratio χ^2^ = 16.694, d.f. = 4; *P* < 0.001) (Figure 2). Most host feeding was by adult female *T. achaeae*. The host feeding in *T. brassicae* was not significantly different from the control group, except when there was competition with other species. In addition, we saw collapsed host eggs that had turned to black only in one replicate of the parasitism treatment (first by *T. achaeae* followed by *T. brassicae*). This shows that larvae of both species had lethal competition. On the other hand, there was no change in color in the rest of the collapsed host eggs.

### 3.2. Functional Response

The *T. brassicae* adult females displayed type I functional responses whereas *T. achaeae* had type II (Figure 4) according to the two statistical analyses carried out, first using Equation (1) and the maximum likelihood (Table 1), and second using Equations (2)–(4), and the lower value of AIC_c_ (Table 2).

### 3.3. Effect of T. achaeae–T. brassicae Interaction on the Functional Response

When there were different densities of parasitized and non-parasitized host eggs, the adult females of both species had type I functional responses (Figure 5); this was determined by the two statistical analyses carried out with the maximum likelihood method using Equation (1) (Table 3) as well as ordinary least square (Equations (2)–(4)) and the lower AIC_c_ values (Table 4).

In the GZLM analysis, the fitted model of dead eggs (by host feeding) was statistically significant (Omnibus test, likelihood ratio χ^2^ = 14.319, d.f. = 1; *P* < 0.001). T. achaeae adult females killed significantly more host eggs by host feeding (avg.: 23.0 ± 2.3%) than did T. brassicae females (avg.: 10.0 ± 2.0%). Furthermore, in the latter species, host egg mortality only occurred at the higher host densities. T. achaeae females preferred killing previously parasitized eggs before non-parasitized ones. This contrasted with the behavior of T. brassicae females, who preferred to feed on non-parasitized eggs (Figure 6).

## 4. Discussion

*Trichogramma* species accepted eggs that have been previously parasitized by the other species (Figure 2 and Figure 3; as along with direct observation). This is consistent with earlier studies on different species of the same genus [49,50], especially under food stress and host scarcity [51]. 

These results can be explained by two effects: (1) that of adult females on previously parasitized hosts (extrinsic competition) as detailed below; or (2) that of interaction between the immature states of both species inside the host (intrinsic competition). Despite the need for further studies to clarify this issue, certain hypotheses can be posited. In relation to extrinsic competition, it should be mentioned that *Trichogramma* species inject protease and phosphatase along with their own egg [52] to disaggregate all the tissues in the host egg, thus making it easier for the parasitoid larva to absorb the host egg quickly [53]. The same behavior has been found in *T. pretiosum* host eggs previously parasitized by *Telenomus heliothidis* Ashmead (Hym.: Scelionidae), so that the immature state of the primary parasitoid is pre-digested in 18 to 20 h [54], and thus exploited to feed *Trichogramma* larvae. Regardless of whether it is extrinsic or intrinsic competition, such an interspecific relationship can be classified as resource-mediated competition (i.e., exploitative competition).

Teder et al. [6] have reported that exploitative competition among any successively acting parasitoids is inevitably strongly asymmetric: the species attacking an earlier developmental stage of the shared host is clearly superior over the parasitoids acting late. However, this does not seem to be the case with the results found in the first trial (Figure 3); when the *T. achaeae* female parasitizes the host first, her offspring are almost predominant. This agrees with Teder et al. [6]. However, when parasitizing second, the emerged adults of *T. achaeae* are significantly similar to the species that parasitized first (Figure 3).

A possible explanation for the findings could be that, when female *T. brassicae* laid eggs after *T. achaeae*, the changes produced by the substances previously injected by *T. achaeae* prevented *T. brassicae* eggs from developing. Vinson and Hegazi [55] also found that the changes in the haemolymph of a parasitized host (used in a larval stage for the purpose of the study) became unfavorable for the development of new eggs of *Campoletis sonorensis* (Cameron) (Hym.: Ichneumonidae). These authors suggested that young embryos in which the embryonic membranes have not yet formed are only able to develop within a narrow range of environments represented by the non-parasitized host. 

Conversely, when female *T. achaeae* laid eggs after female *T. brassicae*, a similar behavior would be expected. However, this did not hamper the embryonic development and later hatching of *T. achaeae* larvae. The only possible hypothesis, in this case, is that the substances injected by *T. brassicae* during oviposition have a lower potential than those injected by the other species. To support such a hypothesis, liquids were extracted from the gasters of two females *Trichogramma* spp.: *T. ostriniae* (Pang et Chen) and *T. dendrolimi* Matsumura; the former were more effective than the latter in preventing the embryonic development of *Ostrinia furcalis* (Guenee) [56]. This proves that there are differences in the substances injected by different *Trichogramma* species during oviposition, which might explain the results in our study.

Regarding intrinsic competition, heterospecific elimination mechanisms have not yet been studied in *Trichogramma* species. However, such mechanisms have been studied in terms of intraspecific competition. Thus, when multiple larvae of the same species are present in the host egg, the initial scramble for food results in the larvae consuming all the egg contents early in development. All the larvae survive if there is sufficient food for all of them to reach a threshold developmental stage; if not, physical proximity results in the attack and consumption of others, continuing until the surviving larvae reach a threshold stage beyond which attacks no longer seem to be effective. The number of larvae remaining at the end of rapid ingestion dictates how many will survive to emerge as adults [57]. The same mechanism is assumed to take place in the *T. achaeae* and *T. brassicae* larvae interaction, in which the larvae of one species are no better competitors than those of the other. This explains the results of the ratio of emerged adults of both species, which was close to 1:1 (Figure 3).

Regarding the second mechanism described above—predation by host feeding—(Figure 2 and Figure 6) a few studies were conducted to determine the host density-dependent model [58]; these include the work by Sahragard et al. [59] and that of Lauziere et al. [60], in which there was no interference competition between parasitoid species. However, until now, there have been no studies on the presence of interference competition, as is the case with ours.

Accepting eggs previously parasitized by another parasitoid species, together with the possibility of eliminating a competitor (whether the competitor species is of the same genus or not), allows *T. achaeae* to be classified as a facultative hyperparasitoid species, as Strand and Vinson [54] determined for *T. pretiosum*. Facultative hyperparasitism is widespread among families of parasitic wasps and probably the most common form of hyperparasitism [61].

Consequently, our results show that, in the presence of previously parasitized host eggs, adult *T. achaeae* females changed their functional response from Type II to Type I (the handling time in this case, *T_h_*, tends to be zero). By contrast, adult female *T. brassicae* maintained a Type I functional response (Figure 4 and Figure 5). The change in the type of functional response in *T. achaeae* relates to changes in *a*′ (instantaneous search rate) and *T_h_* (handling times). As *a*′ is the instantaneous search rate, there are more encounters between the parasitoid and host in the presence of the parasitized host. At the same time, a short handling period (*T_h_*) increases the search time available and hence the likelihood of finding further hosts [46]. The same changes in functional response regarding parasitized prey have been found in IGP predators regarding previously parasitized prey [62].

**Figure 7 insects-15-00496-f007:**
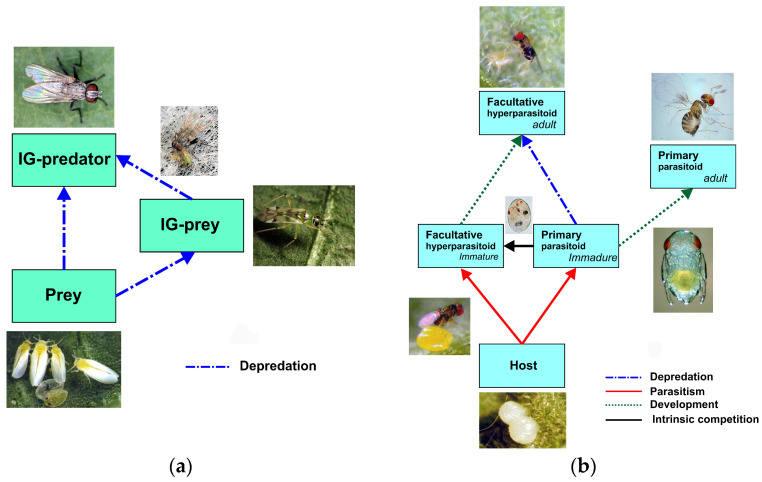
Comparison of intraguild predation (IGP) trophic webs: (**a**) predator–predator IGP (*Coenosia attenuata*–*Nesidiocoris tenuis*–*Bemisia tabaci*) (Ref. [63] and author’s own elaboration), compared to (**b**) parasitoid–parasitoid IGP (*Trichogramma achaeae*–*T. brassicae*–host).

The change in the functional response of *T. achaeae* indicates a change in the parasitism + host-feeding behavior in the better-competing species, which is reflected in a high death rate per capita in the host. This supports the above-mentioned idea that *T. achaeae* is a better competitor than *T. brassicae*. 

Based on our results for *T. achaeae* adult females, the present study provides an insight into the potential coexistence of concurrent (i.e., the parasitoid female uses the same host individual for both feeding and oviposition) and non-destructive host feeding (i.e., the hosts survive the host feeding) in non-parasitized eggs alongside a non-concurrent (i.e., different hosts are used for feeding or oviposition) and destructive host feeding (i.e., the hosts die because of the host feeding) in previously parasitized eggs. These findings warrant further study.

From a theoretical perspective, when it comes to two different prey species and no handling time (*T_h_*), the total opportunism provides maximal energy gain for the predator [64]. Presumably, the same behavior should be displayed by adult female *T. achaeae* in our case, in the presence of two different hosts (parasitized and non-parasitized). The preference observed in adult female *T. achaeae* to feed on host eggs that have been parasitized by the other species results (besides the effect discussed above) in eliminating the immature stage of the competitor (by host feeding), without undergoing loss of fitness; this is achieved by using host eggs that have been previously parasitized by the other species and are thus not feasible for parasitization.

In conclusion, while competition among animals for limited resources is a key factor in ecological relationships, in the case of insect parasitoids, such competition (parasitoid–parasitoid) involves complex relationships, including strict parasitic relationships between the adult and larval stages of different species, as well as predatory interactions between adults and immature stages of the other species.

Interactions between competitors, predators, and their prey have traditionally been viewed as the foundation of community structures. Parasitoids—long ignored in community ecology—are now recognized as playing an important part in influencing species interactions and affecting the ecosystem function [65]. Intraguild predation (IGP) is a dominant interaction in terrestrial food webs that occurs when multiple consumers feed both on each other and on a shared prey (e.g., [66]). 

According to the results obtained here, in the competition between *T. achaeae* and *T. brassicae*, the former species acts as an IG-predator and the latter as an IG-prey in a dual mechanism involving both parasitism and predation. Both are dependent on host density. 

Some hyperparasitoids can attack both the primary parasitoids and the host(s) of these primary parasitoids, and thus act in a way that is functionally identical to intraguild predation. The effect of these facultative hyperparasitoids on biological control is variable, and likely depends on factors such as the attack rate of the facultative hyperparasitoid on the pest versus that of the primary parasitoid [24,67]. 

Our results show the complex relationships between insect parasitoid species (Figure 7b), in particular, the parasitoid–parasitoid IGP compared to the predator–predator IGP shown in Figure 7a [phytophagous prey *Bemisia tabaci* (Gennadius) (Hem.: Aleyrodidade), IG-predator *Coenosia attenuata* Stein (Dip.: Muscidae) y IG-prey *Nesidiocoris tenuis* (Reuter) (Hem.: Miridae)]. As a consequence of the above results, two future research lines may arise as follows:-From an experimental standpoint, the feeding of adult females *T. achaeae*, besides eliminating competitors, involves a fitness gain in terms of increasing female fertility and/or longevity by feeding on host eggs previously parasitized by other species, compared to feeding on non-parasitized host eggs or other non-host foods (e.g., nectar and/or pollen).-From a theoretical standpoint, mathematical models on this subject have mainly considered facultative hyperparasitism (e.g., [68]) and host feeding (e.g., [69,70]) separately. An exception to this is Borer’s work [71], in which both aspects are considered, although he does not take into account the effects of host feeding on the direct mortality of IG prey (the primary parasitoid). The results found in the present study can go towards developing a more realistic model.

## Figures and Tables

**Figure 1 insects-15-00496-f001:**
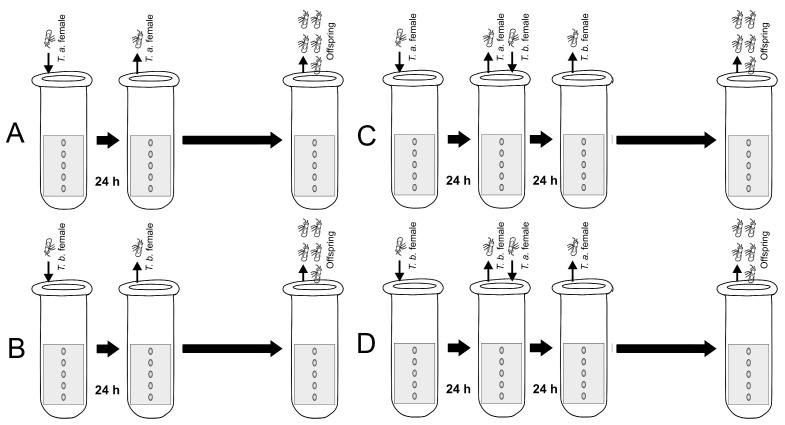
Diagram showing the treatments carried out with *Trichogramma achaeae* (*T.a*.) and *T. brassicae* (*T.b.*): (**A**) female *T.a.* parasitizing unparasitized hosts, (**B**) female *T.b.* parasitizing unparasitized hosts, (**C**) female *T.b.* parasitizes eggs previously parasitized by *T.a.*, and (**D**) female *T.a.* parasitizes eggs previously parasitized by *T.b*.

**Figure 2 insects-15-00496-f002:**
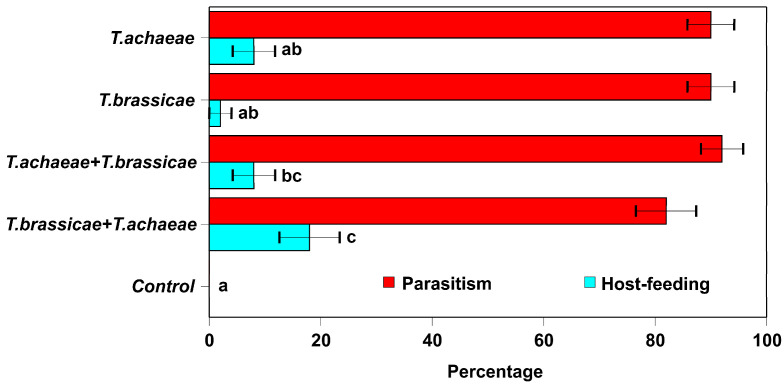
Dead *E. kuehniella* egg percentage (±SE) resulting from the activity of the *T. achaeae* and *T. brassicae* females (parasitism or host feeding) in competition with the other species, under laboratory conditions (25 ± 1 °C, 60–80% RH, and 16:8 h L:D cycle) (bars followed by different letters indicate significant differences at *P* = 0.05).

**Figure 3 insects-15-00496-f003:**
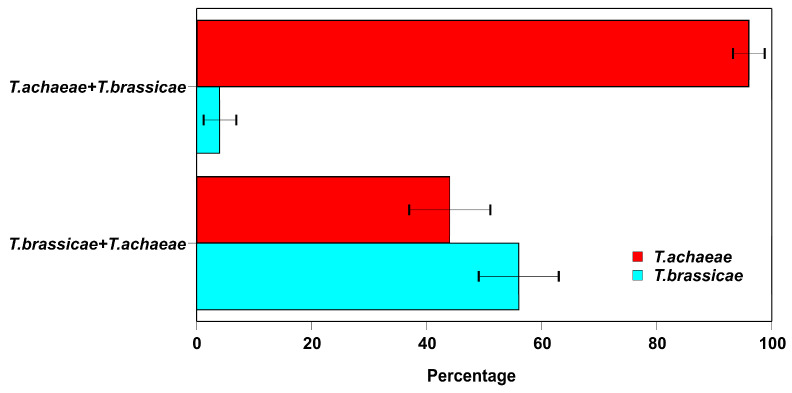
Percentage (±SE) of parasitoid adult emergency from *E. kuehniella* host eggs parasitized by *T. achaeae* and *T. brassicae* female adult, in competition with other species, under lab conditions (25 ± 1 °C, 60–80% RH, and 16:8 h L:D cycles).

**Figure 4 insects-15-00496-f004:**
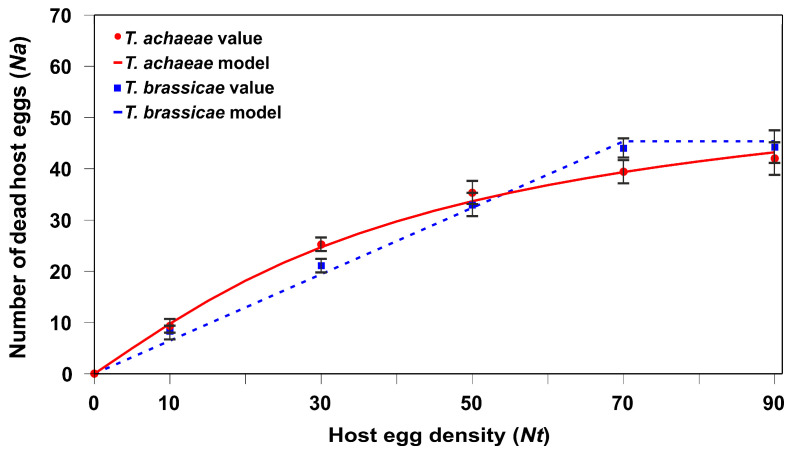
Mean (±SE) number of dead *E. kuehniella* eggs (parasitism + host feeding) resulting from the parasitic activity of *T. achaeae* or *T. brassicae* female adult, and the predicted values according to the functional response at different density levels under lab conditions (25 ± 1 °C, 60–80% RH, and 16:8 h L:D cycles).

**Figure 5 insects-15-00496-f005:**
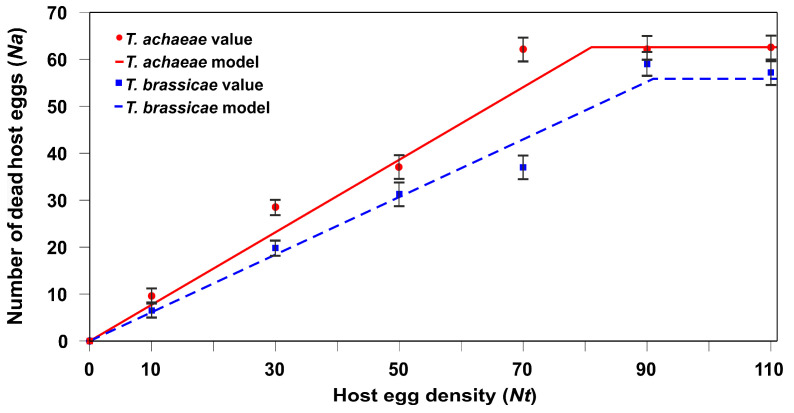
Mean (±SE) number of dead *E. kuehniella* eggs (parasitism + host feeding) resulting from the parasitic activity of *T. achaeae* and *T. brassicae* female adults, along with the predicted values according to the functional response at different host density levels, in competition with the other species, under laboratory conditions (25 ± 1 °C, 60–80% RH, and 16:8 h L:D cycles).

**Figure 6 insects-15-00496-f006:**
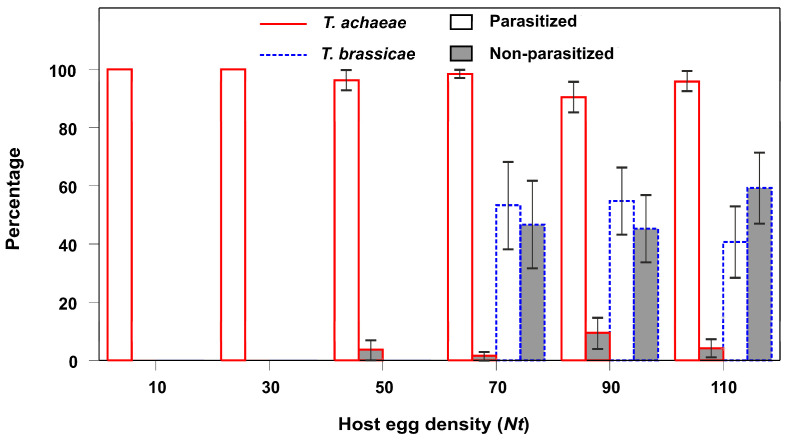
Relative percentage (±SE) of *E. kuehniella* eggs killed by host feeding alone, resulting from the activity of *T. achaeae* and *T. brassicae* adult female, according to the host density and whether they had previously been parasitized by the other species, under laboratory conditions (25 ± 1 °C, 60–80% RH, and a 16:8 h L:D cycle).

**Table 1 insects-15-00496-t001:** Selection of the functional response model based on the analysis of the maximum likelihood parameter estimates (±SE) using the proportion of hosts killed (parasitism + host feeding) as a polynomial function of host density resulting from the parasitic activity of *T. achaeae* and *T. brassicae* female under laboratory conditions (25 ± 1 °C, 60–80% RH, and 16:8 h L:D cycles).

Species	Parameters	Values (±SE)	Confidence Interval (*P* = 0.05)		Best-Fit Model
			Lower Limit	Upper Limit	
*T. achaeae*	*P*_0_ (interception)	3.506 ± 0.571	−3.745	10.758	Type II
	*P*_1_ (linear)	−0.477 ± 0.036	−0.938	−0.017	
*T. brassicae*	*P*_0_ (interception)	1.660 ± 1.1250	−3.181	6.501	Type I
	*P*_1_ (linear)	−0.022 ± 0.050	−0.235	0.191	

**Table 2 insects-15-00496-t002:** Parameters and statistical significance of the functional response equations for the number of dead *E. kuehniella* eggs (parasitism + host feeding) resulting from the parasitic activity of *T. achaeae* and *T. brassicae* under laboratory conditions (25 ± 1 °C, 60–80% RH, and a 16:8 h L:D cycle) (*a*′ = instantaneous search rate, days^−1^; *α* = parasitoid mortality potential; and *T_h_* = handling time, days).

Species	Functional Response	Parameters (±SE)		Statistical Parameters		
		*a*′/α (day^−1^)/(per-unit)	*T_h_* (day)	d.f.	*R* ^2^	AIC_C_
*T. achaeae*	Type I	0.809 ± 0.113	—	5	0.859	2.564
	Type II	9.996 ± 4.973	0.018 ± 0.001	5	0.998	2.336
	Type III	64.830 ± 22.355	0.018 ± 0.003	5	0.997	3.463
*T. brassicae*	Type I	1.043 ± 0.044	—	5	0.994	2.564
	Type II	2.653 ± 1.220	0.011 ± 0.003	5	0.981	8.608
	Type III	0.137 ± 0.064	0.015 ± 0.001	5	0.982	8.417

**Table 3 insects-15-00496-t003:** Selection of the functional response model based on analysis of the maximum likelihood parameter estimates (±SE) using the proportion of hosts killed (parasitism + host feeding) as a polynomial function of the host density resulting from the parasitic activity of *T. achaeae* and *T. brassicae* females, in competition with the other species, under laboratory conditions (25 ± 1 °C, 60–80% RH, and a 16:8 h L:D cycle).

Species	Parameters	Values (±SE)	Confidence Interval (*P* = 0.05)		Best-Fit Model
			Lower Limit	Upper Limit	
*T. achaeae*	*P*_0_ (interception)	3.586 ± 2.010	−5.046	12.218	Type I
	*P*_1_ (linear)	−0.047 ± 0.077	−0.380	0.285	
*T. brassicae*	*P*_0_ (interception)	0.024 ± 0.402	−5.133	5.085	Type I
	*P*_1_ (linear)	0.075 ± 0.028	−0.282	0.431	

**Table 4 insects-15-00496-t004:** Parameters and statistical significance of the functional response equations for the number of dead *E. kuehniella* eggs (parasitism + host feeding) resulting from the parasitic activity of *T. achaeae* and *T. brassicae* females in competition with the other species, under laboratory conditions (25 ± 1 °C, 60–80% RH, and 16:8 h L:D cycles) (*a*′ = instantaneous search rate, days^−1^; *α* = parasitoid mortality potential; and *T_h_* = handling time, days).

Species	Functional Response	Parameters (±SE)		Statistical Parameters		
		*a*′/α (day^−1^)/(per-unit)	*T_h_* (Day)	d.f.	*R* ^2^	AIC_C_
*T. achaeae*	Type I	0.879 ± 0.072	—	5	0.949	8.658
	Type II	1.652 ± 0.789	0.007 ± 0.002	5	0.960	10.072
	Type III	0.121 ± 0.011	0.014 ± 0.001	5	0.935	12.439
*T. brassicae*	Type I	0.951 ± 0.065	—	5	0.972	7.601
	Type II	0.805 ± 0.419	0.003 ± 0.001	5	0.965	11.322
	Type III	0.034 ± 0.018	0.008 ± 0.002	5	0.932	13.341

## Data Availability

The raw data supporting the conclusions of this article will be made available by the authors on request.

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
