# Peer review of "Importance of Host Feeding in the Biological Control of Insect Pests: Case Study of Egg Parasitoid Species (Hymenoptera: Chalcidoidea: Trichogrammatidae)"

_insects, 2024, doi:10.3390/insects15070496_

Round 1

Reviewer 1 Report

Comments and Suggestions for Authors

This manuscript deals with a very interesting and poorly understood, and has a great potential, but the way it is actually written and structured makes it hard to follow and understand.

To begin with, the title is confusing. At first I though the manuscript was addressing how the food provided to the host can influence biological control, not the influence of how parasitoids feed on the hosts. And honestly, I do not think (once I understand the true meaning) that is the main subject in the manuscript.

I recommend to make a substantial restructuring and a thorough revision of the English grammar and spelling. Next I will explain my concerns and suggestions, section by section.

Abstract

The abstract is hard to follow. It is very difficult to understand without having read the whole manuscript. Some terms are used without explanation, such as type II or instantaneous search rate, and therefore the reader can loose interest in the paper.

Some minor comments:

Line 23: “guild” appears twice, I suggest to rephrase the sentence: “Intraguild predation in parasitoids has drawn special attention…”

Line 25: “Trichogramma” should be in italics.

Lines 27-28: “The results show the importance of host-feeding”, this sentence looks unrelated.

Lines 36-38: These two last sentences are empty, I suggest to remove them and focus on the main results and their implications.

Introduction

Introduction is a little messy, it needs to be revised in order to help the reader to understand why this research is needed and which is the knowledge gap. Here I give some suggestions that I think can help:

Line 49: why are you suddenly including predators when you are only talking about parasitoids?

Lines 53-54: what do you mean? Which “dramatic case of potential damage” are you referring at?

Line 58: what do you mean with “a patch that has been previously parasitized”? Which kind of patch?

Lines 64-65: So intrinsic competition occurs in superparasitism/multiparasitism cases, and even facultative hyperparasitism, which has been explained within the extrinsic competition. This needs to be better organized.

The whole 68-78 paragraph looks unnecessary: the only intraguild competition that you need to focus on is the one you already explained in the previous paragraph.

Line 84: Not all parasitoids are synovigenic. Maybe you are talking about some specific parasitoid groups. Please revise it.

Materials and Methods

Lines 128, 204 and 207: “Trichogramma” should be in italics.

Lines 134-136: Please mention which laboratory you obtained the cultures from, and do not repeat it in lines 142-143, nor in experiments 2 and 3. Actually, there is some general information, common to all experiments, such as the lab conditions, the number of repetitions, and others.

Lines 138-139: What do you mean with “one factor at five treatment levels”? Which is the factor? And according to figure 1, you have 4 treatments; I guess that you are including the control treatment, but in experiment 2 you mentioned five treatments, without including the control one.

Subsection headings should be in concordance with the content. In 2.1 you do explain where the material came from, but I do not think the experiments should be within the “biological material” section.

Line 146: Why do you need to determine sex if you are ensuring male offspring?

Figure 1 might be confusing since treatment C is very close to treatment A, as treatment D is close to B. In this way, it looks like C and D are continuations of A and B, respectively. Also, there is only a long arrow in treatments A and B; I guess you are representing the offspring emergence, but then place a similar arrow in treatments C and D.

Lines 229-233: I do not understand where you got the instantaneous search rate and the handling time from.

Results

Line 267: “Trichogramma” should be in italics.

Line 268: Are you saying that the check group showed differences? Please revise this sentence. A similar confusing sentence is in line 288 (“host-feeding was not significantly different from the check group except when there was no competition with other species”, but no significant relationship is seen in the figure).

Line 286: Considering that you are talking about the eggs being parasitized, it seems that you should be referring to figure 2 instead of figure 3. Therefore, nothing is said about figure 3, which is showing parasitoid emergency.

Line 294-295: The sentence is confusing, please revise it.

Discussion

Discussion is somewhat hard to follow, because of the writing and because things like “results from figure 3” are said instead of showing the actual results. Furthermore, some of the results receive a lot of attention, such as those from figure 3 (of which paradoxically nothing was said in results), but others lack, such as how host density is affecting parasitoidism and competition. Also, despite being in the title, host-feeding is only shortly discussed. Additionally, figure looks unnecessary, and I feel that the implications of the results in biological control are not clearly discussed.

Other minor comments:

Line 388: “Trichogramma” should be in italics.

Line 415: “T. achaeae” should be in italics.

Comments on the Quality of English Language

English language should be revised (Ex, lines 324-325: Then adult female of T. achaeae kills by host-feeding more significant host eggs than females of T. brassicae”; lines 327-328: “it includes the distribution of dead host eggs by host-feeding according to them were or not previously parasitized by the other species”; line 330: “Different was the behavior of T. brassicae”).

Author Response

Thank you very much for your comments which have been taken into account in the revised version of the MS. Please see attached PDF file.

Reviewer 2 Report

Comments and Suggestions for Authors

The work is interesting and articulated, and only some clarification is asked. I have attached files with suggestions listed by the manuscript line number.

Author Response

(The authors gave the same response as above.)

Round 2

Reviewer 1 Report

Comments and Suggestions for Authors

I can see that this version has substantially been improved, but there are still some issues, many of them are an apparent result of a rushed correction without a last-minute revision. There are many mistakes, many of them I am pointing out, but there are many others that should be revised (mainly punctuation marks, or misplaced brackets).

I still think the title should be clearer, maybe just by placing a hyphen between “host” and “feeding”, like it is in the whole document, it would be enough.

In discussion please do not refer to figures but to the results themselves. And since you have decided to keep the Figure 7, you should remove the 7a diagram, because it is not derived from your results. If you still want to keep it, you should indicate in the figure caption where do these results come from.

More specific comments:

Line 25: “extrinsic”.

Line 27: “The results show the importance of of host-feeding”, this sentence still seems unrelated.

Lines 37-38: Please revise the excess of brackets.

Lines 52-53: I still think that there is no need of mentioning from nowhere the predators. The sentence could be “Besides the host range width, parasitoid guilds can be defined by the developmental stage at which the host is attacked by the parasitoid and the mode of parasitism (i.e., ectoparasitic o endoparasitic) ([2]; [4])”.

Line 61: You have already mentioned intrinsic and extrinsic competition in the previous paragraph. I suggest to continue the paragraph in line 60: “Therefore, extrinsic competition among free-living parasitoid adults searching for host resources) occurs when…”

Line 73: This is the first time you mention the acronym IGP within the text (the abstract does not count), so you need to indicate what it means.

Line 131: Again, I do not think that the experiments should be included within the biological material section. Experiments should be in a separate section, such as 2.2 Experimental design, for instance.

Line 140: “in which”.

Line 201: “by using”, “in rows of 3x1-cm cardboards”.

Line 202: “The number of repetitions”.

Line 204: “Trichogramma” in italics.

Line 205: “The adults were removed afterwards”.

Line 206: “to evolve until the emergence”.

Line 209: “simultaneously over the test period”.

Line 225: “to the three functional response types”.

Lines 241-242: “Tablecurve 2D” is twice.

Lines 246-247: please revise the sentence, and also the two levels would be clear, not one of them between brackets.

Line 249: one of two, “previous” or “prior”, and “in” or “at” should be kept. This is a mistake that is present many times along the manuscript, please revise it thoroughly.

Line 294: You can not be referring to Fig. 3, please revise it. Or is this entire paragraph useless? It seems that these results are already explained in page 10, lines 330-339.

Line 334: What do you mean with “the higher host”? Maybe you mean “the highest host density”?

Line 361: The first effect should be clearer. Which is the effect from adult females? Only by saying “adult females”, the effect can not be explained, as you say in that sentence.

Line 367: “to absorb”.

Line 371: It is not an intraspecific relationship but interespecific.

Line 377: According to what you have explained before, it does seem exactly the case of your results. Please revise it.

Line 431: “further hosts”.

Line 439: “behaviour”.

Line 449: “prey species”.

Line 464: “parasitoids” (instead of parasites).

Lines 480-481: There are some Spanish words in the text, please revise it.

Author Response

Please, see in the attached PDF file.
